# Sarcopenic Obesity in Liver Cirrhosis: Possible Mechanism and Clinical Impact

**DOI:** 10.3390/ijms22041917

**Published:** 2021-02-15

**Authors:** Hiroki Nishikawa, Hirayuki Enomoto, Shuhei Nishiguchi, Hiroko Iijima

**Affiliations:** 1Department of Internal Medicine, Division of Gastroenterology and Hepatology, Hyogo College of Medicine, Nishinomiya, Hyogo 663-8501, Japan; enomoto@hyo-med.ac.jp (H.E.); hiroko-i@hyo-med.ac.jp (H.I.); 2Center for Clinical Research and Education, Hyogo College of Medicine, Nishinomiya, Hyogo 663-8501, Japan; 3Department of Internal Medicine, Kano General Hospital, Osaka 531-0041, Japan; nishiguchi@heartfull.or.jp

**Keywords:** liver cirrhosis, sarcopenic obesity, mechanism, clinical impact

## Abstract

The picture of chronic liver diseases (CLDs) has changed considerably in recent years. One of them is the increase of non-alcoholic fatty liver disease. More and more CLD patients, even those with liver cirrhosis (LC), tend to be presenting with obesity these days. The annual rate of muscle loss increases with worsening liver reserve, and thus LC patients are more likely to complicate with sarcopenia. LC is also characterized by protein-energy malnutrition (PEM). Since the PEM in LC can be invariable, the patients probably present with sarcopenic obesity (Sa-O), which involves both sarcopenia and obesity. Currently, there is no mention of Sa-O in the guidelines; however, the rapidly increasing prevalence and poorer clinical consequences of Sa-O are recognized as an important public health problem, and the diagnostic value of Sa-O is expected to increase in the future. Sa-O involves a complex interplay of physiological mechanisms, including increased inflammatory cytokines, oxidative stress, insulin resistance, hormonal disorders, and decline of physical activity. The pathogenesis of Sa-O in LC is diverse, with a lot of perturbations in the muscle–liver–adipose tissue axis. Here, we overview the current knowledge of Sa-O, especially focusing on LC.

## 1. Sarcopenia and Sarcopenic Obesity in Liver Cirrhosis

Sarcopenia refers to the loss of skeletal muscle mass (SMM) and muscle strength or physical function [1]. Sarcopenia is rapidly evolving in the clinical research fields, especially in the last few years, after the proposal of its concept by Irwin Rosenberg in 1989 [2]. Primary sarcopenia is defined as a condition in which SMM and muscle strength or physical function decline with aging, while secondary sarcopenia is defined as a condition in which SMM and muscle strength or physical function are impaired due to underlying diseases [3,4]. In recent years, a large amount of evidence on sarcopenia has been accumulated, and the disease has been registered in ICD-10, and sarcopenia is now recognized as a disease rather than a concept.

Liver cirrhosis (LC) is the end-stage form of chronic liver disease (CLD) and presents with a variety of complications. Liver cirrhosis (LC) is a representative disease that causes secondary sarcopenia [5]. LC leads to secondary sarcopenia, due to characteristic conditions such as protein-energy malnutrition (PEM). In cirrhotic patients, fasting for only 6–10 h is equivalent to fasting for 2–3 days in healthy individuals, which is also associated with the development of sarcopenia [6]. In LC patients, there is an amino acid imbalance with the decrease of branched-chain amino acids (BCAAs) and increase of aromatic amino acids [5]. BCAAs are amino acids that are metabolized by muscles, to detoxify ammonia, and are a good source of energy for the liver [5]. BCAAs were found to improve homeostasis model assessment of insulin resistance (HOMA-IR) scores and the function of beta cells in CLD patients [7]. The complication rate of sarcopenia in LC patients is reported to be 30–70%, which is obviously high, considering that the complication rate of sarcopenia in inflammatory bowel disease (a typical disease that causes secondary sarcopenia) is about 20% [8]. This fact may be related to both the fact that cirrhosis is likely to be complicated with secondary sarcopenia and the fact that Japanese LC patients are aging. Direct-acting antivirals can almost eliminate the virus in patients with cirrhosis C, and nucleoside analogues can also control the disease in patients with cirrhosis B. As a result, the survival has significantly improved, but at the same time, Japanese LC patients have become older. In cirrhotic patients, the annual rate of SMM loss was reported to be 1.3% in Child-Pugh A, 3.5% in Child-Pugh B, and 6.1% in Child-Pugh C [9]. The annual rate of muscle loss increases with worsening liver reserve, which is clearly higher than the 1% annual rate of muscle loss in the general elderly population [9].

Sarcopenic obesity (Sa-O) was first defined by Baumgartner as the co-existence of sarcopenia and obesity, as measured by dual-energy X-ray absorptiometry, describing an interplay between obesity and sarcopenia related to physical activity decline and reduced energy expenditure [10]. The rapidly increasing prevalence and poor clinical consequences of Sa-O are recognized as an important public health problem in the aging population. The main alterations in body composition due to aging include an increase in body fat and a decrease in SMM, which are not accompanied by much change in BMI [11]. These alterations of body composition are often difficult to notice on the outside, and detection tends to be delayed, making it easier for lifestyle-related diseases and other conditions to progress without being noticed. The picture of CLDs has changed considerably in recent years. One of them is the increase of non-alcoholic fatty liver disease (NAFLD). In Asia, NAFLD prevalence increased significantly over time (25.3% between 1999 and 2005, 28.5% between 2006 and 2011, and 33.9% between 2012 and 2017; *p* < 0.0001) [12]. More and more CLD patients, even those with LC, tend to be presenting with obesity these days, due to the changes of lifestyle. As PEM in LC can be invariable, the patients probably present with Sa-O. In most cases, sarcopenia and obesity are judged separately, and Sa-O is considered when both are present. However, the biggest problem in clinical studies of Sa-O is the lack of a common definition, despite the fact that Asian Working Group for Sarcopenia (AWGS) and European Working Group on Sarcopenia in Older People (EWGSOP) recommend a definition of sarcopenia [3,4]. The indexes of obesity and the definition of obesity are not consistent among studies, and thus the frequency of Sa-O varies widely.

## 2. Obesity Paradox and Liver Cirrhosis

Obesity can be a major risk factor for morbidity and mortality in metabolic and cardiovascular diseases and is an important health threat. WHO proposes the definition of obesity according to body mass index (BMI): ≥30 kg/m^2^ in Caucasians, and ≥25 kg/m^2^ in Asians, with further classifications into class I, II, and III obesity [13,14]. The prevalence of obesity in adults has doubled since 1980 and globally continues to elevate [11]. Obesity can accelerate the progression of LC status, but the data are not consistent on the extent to which it affects mortality. This discrepancy is partly due to the lack of correction for body weight for excess body water and partly due to the obesity paradox [15,16,17]. Obesity paradox is a phenomenon in which obese people are found to have a reduced risk of death compared to people with standard weight, and it has attracted attention in considering the pathological significance of obesity. Karagozian et al. reported, in their large cohort study (32,605 LC subjects), that crude mortality was lower for obese LC subjects than for non-obese LC subjects (2.7% vs. 3.5%, *p* = 0.02), and in their multivariate analysis, obesity significantly lowered a risk of inpatient mortality (hazard ratio (HR) = 0.73, *p* = 0.02) [16]. Most studies on obesity paradox use BMI as an index of obesity, but obesity is essentially a condition associated with increased adipose tissue mass, and it is difficult to accurately assess adipose tissue mass with BMI, which includes muscle mass and bone mass in addition to fat. There are cases of excess visceral fat even when BMI is within the normal range, and attention should also be paid to the amount of visceral fat in cases with sarcopenia.

## 3. Prevalence and Definition of Sarcopenic Obesity

Sarcopenia and obesity are closely related. Sa-O involves a complex interplay of physiological mechanisms, including increased inflammatory cytokines, oxidative stress, insulin resistance, hormonal disorders, and decline of physical activity [10,11,18,19]. However, as mentioned above, there is no mention of Sa-O in either the AWGS guidelines or the EWGSOP guidelines. Existing LC-related reports are limited, but they have shown that a Sa-O prevalence in cirrhotic patients is 2% to 42% [20,21,22,23,24,25,26,27,28,29]. The definition and prevalence of Sa-O in LC patients are summarized in Table 1. The most commonly used definition of Sa-O is the combined assessment of skeletal muscle index (SMI, SMM measured by computed tomography (CT, L3 level) corrected for height squared) and BMI (>25 or 30 kg/m^2^), and in most LC-related studies regarding Sa-O, grip strength and walking speed are not incorporated in the definition. However, there are no internationally standardized diagnostic criteria for Sa-O. This may be, in part, due to differences in BMI and SMM among races.

## 4. Molecular Mechanism and Liver Diseases

Various molecular biological findings and clinical analyses have been reported on the mechanisms of metabolic abnormalities associated with LC. The BMI of LC patients in Japan is almost the same as that of general population of the same age, and one in four LC patients is obese with a BMI ≥ 25 kg/m^2^, and only a few LC patients have a BMI < 18.5 kg/m^2^ [30]. In our 226 Japanese LC patients, 70 patients (31%) had a BMI ≥ 25 kg/m^2^, and 14 patients (6%) a BMI < 18.5 kg/m^2^.

Clinical features of glucose metabolism in LC patients are as follows: (1) not-so-higher fasting blood glucose in the early morning, (2) higher blood insulin concentration, (3) higher HOMA-IR level in more than half of LC patients, and (4) postprandial hyperglycemia. In addition, hepatitis C virus infection, which is the most common cause of cirrhosis in Japan, can be a cause of increased hepatic steatosis and insulin resistance through the action of hepatitis C virus core protein [31,32]. There are many reports that insulin resistance and obesity are important risk factors for carcinogenesis in cirrhotic patients [33]. The pathogenesis of Sa-O in LC is diverse, with a lot of perturbations in the muscle-liver–adipose tissue axis.

Insulin has metabolic effects that promote anabolism and inhibit catabolism of nutrients. As skeletal muscle is the target organ of insulin, loss of SMM is a cause of insulin resistance [34]. Insulin resistance in skeletal muscle is involved in the pathogenesis of sarcopenia and may contribute to obesity [34]. Sarcopenia can increase the risk of liver fibrosis progression in patients with obesity, insulin resistance, metabolic syndrome, and liver steatosis, which can easily fall into Sa-O [35,36,37]. LC patients are likely to be involved in insulin resistance and chronic inflammation [38]. Oxidative stress is thought to be a mechanism linking insulin resistance and inflammation to sarcopenia. Oxidative stress can damage mitochondria and nuclear DNA, stimulate apoptosis, and lead to muscle fiber atrophy and myocyte loss [39]. Dysbiosis indicates a state where the diversity of gut microbiota and the number of bacteria in the gut are reduced [5]. LC associated dysbiosis also induces oxidative stress [5]. Testosterone, which is important for skeletal muscle formation, is decreased in LC patients [40]. A previous randomized trial emphasized the significance of testosterone supplementation therapy on the improvement of SMM in male LC patients [40].

Adipose tissue produces inflammatory proteins and cytokines, such as CRP, TNF-α, IL-6, and IL1β, forming a chronic inflammatory environment that leads to muscle atrophy and sarcopenia [41]. Adipose tissue produces a number of hormones and bioactive substances, including leptin and adiponectin. Adiponectin is present in large amounts in the blood, declines with fat accumulation, and increases with weight loss. Adiponectin has the ability to improve insulin sensitivity throughout the body. Adiponectin stimulates the AMPK pathway and suppresses the NF-κB pathway. It also suppresses the production of TNF-α and IFN-γ secreted by monocytes, macrophages, and dendritic cells, and it increases anti-inflammatory cytokines. TNF-*α* and IFN-s-dependent muscle decline is linked to NF-κB pathway [42]. The AMPK stimulation of adiponectin is attenuated in obesity [41]. The obesity-induced suppression of adiponectin leads to persistent chronic inflammation in muscle tissue [41]. Especially in patients with NAFLD, the protective effects of adiponectin for hepatocytes have been widely examined [43,44]. Adiponectin exerts an anti-fibrotic and anti-inflammatory effect in LC [45]. A previous prospective study reported that adiponectin was associated with the degree of liver decompensation and worse prognosis in LC patients [46]. Adiponectin levels are reported to be independently associated with sarcopenia in LC patients [47]. Leptin, on the other hand, reflects whole-body fat mass. Leptin is an adipokine that regulates energy balance and glucose homeostasis. Obesity causes leptin resistance, which is thought to be indirectly related to sarcopenia through insulin resistance and other mechanisms [41]. In obesity-associated fatty liver, leptin induces CD14 expression via activation of STAT3 signaling in Kupffer cells, leading to marked inflammation and fibrosis by increasing the responsiveness of the liver to low dose of endotoxin [48].

Ectopic fat accumulation and SMM decline may exist in a vicious cycle due to their mutual influence [18]. Sarcopenia reduces physical activity, leading to energy expenditure loss and the elevated risk of obesity [19]. In contrast, increase of a visceral fat induces inflammation, contributing to the sarcopenia incidence [49]. Kim et al. reported, in a longitudinal study, that visceral obesity was associated with future SMM decline in Korean individuals [50]. In male cirrhotic patients with hepatocellular carcinoma (HCC) undergoing liver transplantation, a visceral adipose tissue ≥ 65 cm^2^/m^2^ raised the risk of HCC recurrence more than five times [51]. Accumulation of subcutaneous adipose tissue can be an adverse predictor in patients with alcoholic LC [52]. High subcutaneous adipose tissue density and high visceral adipose tissue density on CT significantly correlated negatively with survival in HCC patients [53].

It has been widely accepted that the inhibition of myostatin contributes to reduced adipose tissue, indicating myostatin acts either directly or indirectly on adipose tissue [54]. Myostatin also has the effect of inhibition of protein synthesis in skeletal muscle [55]. Increased myostatin levels have been associated with both obesity and insulin resistance [56,57,58]. Serum myostatin levels can be lowered by aerobic exercise [58]. Resistance training can reduce myostatin levels, leading to the improvement of insulin resistance [59]. Hittel et al. reported that extremely obese individuals have been reported to have a 35% increase in plasma mature myostatin level and a 23% increase in skeletal muscle precursor myostatin level, as compared to lean controls, and BMI strongly correlated with myostatin level in skeletal muscle [56]. Visceral fat area, HbA1c, myostatin, and leptin are reported to be contributing factors associated with increased liver fat accumulation in patients with non-obese NAFLD [60]. Ammonia is a cytotoxic substance produced via several physiological processes, such as amino acid catabolism and gut microbiota metabolism [61]. The hepatocyte is the only cell that is able to metabolize ammonia to urea, which is a nontoxic metabolite being excreted by the kidneys. The enzymes that catalyze the urea cycle are zinc-containing enzymes, and when zinc deficiency occurs, ammonia processing is reduced. Most LC patients have zinc deficiency [62]. In LC patients, due to liver dysfunction and portosystemic shunts, serum ammonia concentrations tend to increase [56]. In our previous data in LC patients, serum myostatin level significantly increased with the worsening of liver function and the increase of serum ammonia level, and was associated with clinical outcomes [55]. Higher serum ammonia level can cause higher myostatin level in skeletal muscle, resulting in the progression of sarcopenia through the suppression of muscle protein synthesis [55].

Growth hormone (GH), as well as insulin-like growth factor (IGF)-1, regulates the growth of tissues, including skeletal muscle. Fatty acids inhibit the production of GH and decrease IGF-1 levels. The anabolic hormone IGF-1 is also involved in the regulation of GH, and IGF-1 acts in response to GH. IGF-1 decreases with increasing fat mass. In obese individuals, these anabolic hormones are decreased, and these decreases may be related to skeletal muscle disorders. LC is a condition of acquired resistance to GH. Impaired GH-IGF-1 system in LC can lead to LC-related complications [63]. In LC patients, IGF-1 level was correlated with the degree of liver dysfunction [63].

## 5. Myosteatosis and Liver Diseases

Myosteatosis refers to the pathological fat accumulation in skeletal muscle within the muscular fibers or within the fascia of the skeletal muscle [64]. Myosteatosis is associated with insulin resistance and type 2 diabetes mellitus [65]. Myosteatosis meditates inflammatory responses and can be linked to decreased muscle function and SMM decline caused by muscle atrophy and physical disabilities [66,67].

Myosteatosis in LC patients increased the risk of death by 1.5-to 2-fold, mainly due to the higher incidence of sepsis-related death [20]. Kaibori et al., reported, in their 141 HCC patients undergoing hepatectomy, that the five-year overall survival (OS) rates in patients with and without higher intramuscular adipose tissue content (IMAC) were 46% and 75%, and the five-year disease-free survival rates in patients with and without higher IMAC were 18 and 38%, and that higher IMAC was an independent predictor for OS in their multivariate analysis [21]. They also reported the significant correlation between higher IMAC and liver dysfunction, a higher amount of intraoperative blood loss, and complications of diabetes mellitus [21]. Another Japanese study reported the close correlation between higher IMAC and bacteremia after liver transplantation [68]. A significant correlation between preoperative higher IMAC and pulmonary dysfunction after hepatectomy was also reported [69]. Meanwhile, Bhanji et al. reported, in their LC subjects (*n* = 675), that myosteatosis was identified in 348 patients (52%), sarcopenia in 242 (36%), and hepatic encephalopathy in 128 (19%), and in their multivariable analysis, both myosteatosis and sarcopenia were independent factors for hepatic encephalopathy (HR = 2.25 and 2.42; *p*-values both < 0.01) [70].

## 6. Dysbiosis and Sarcopenic Obesity in Liver Cirrhosis

Humans live in symbiosis with about 100 trillion intestinal bacteria of about 1000 species. The gut microbiota is closely related to the nutritional metabolism (lipids, energy, etc.) and immune function of the human host. About 10% of the daily energy requirement of humans is provided by the fermentation of intestinal bacteria in the colon [5]. Previous studies have shown an association between LC and the gut microbiome. Li et al. performed a whole microbiome analysis on fecal samples collected from 98 cirrhotic patients and 83 healthy controls [71]. Their quantitative metagenomics analysis revealed 75,245 genes that differed markedly in abundance between the two groups, and most of these genes were classified into 66 clusters, representing homologous bacterial species. Among these, 28 bacterial species were found to be abundant in cirrhotic patients, mostly of oral origin (mainly *Veillonella* and *Streptococcus*) [71]. Harmful bacteria can cause hyperammonemia by the decomposition of urea, leading to hypermyostatinemia in skeletal muscle and sarcopenia [5].

The significant correlation between obesity and gut dysbiosis has been attracting much attention. Obesity induces dysbiosis and increased production of lipopolysaccharide (LPS) and intestinal permeability [72,73]. LPS causes inflammatory skeletal muscle damage through the gut–liver–muscle axis and is involved in the formation of sarcopenia [74]. Liu et al. reported that obesity-related gut microbial species were associated with changes in circulating metabolites. In their data, the significant decrease of abundance of *Bacteroides thetaiotaomicron* (glutamate-fermenting bacteria) in obese subjects was observed, and bariatric surgery for weight reduction partially reversed obesity-associated microbial alterations and metabolic alterations (i.e., elevated serum glutamate level) in obese subjects [75]. Dysbiosis in LC can lead to (a) reduced bacterial diversity [76,77], (b) reduced short chain fatty acid [78], (c) tight junction failure and increased intestinal permeability [79], (d) decreased antioxidant effect [80], and (e) endotoxemia [81,82]. These can be linked to anabolic resistance, chronic inflammation, decreased mitochondrial function, oxidative stress, and insulin resistance, resulting in LC progression and sarcopenia. In view of these, dysbiosis appears to be closely linked to Sa-O in LC. The potential interplay among patients with sarcopenia, liver cirrhosis, obesity, myosteatosis, and dysbiosis is demonstrated in Figure 1.

## 7. Prognosis and Intervention in Cirrhotic Patients with Sarcopenic Obesity

Vidot et al. reported, in cirrhotic patients undergoing liver transplantation (*n* = 205), that all men with obesity classified as well-nourished status had sarcopenia, and 62% of women with obesity classified as well-nourished status had sarcopenia, and that serum testosterone decline was an independent predictor for muscle wasting [29]. Montano-Loza et al. reported, in their 678 LC patients, that patients with sarcopenia (median OS, 22 months), Sa-O (median OS, 22 months), and myosteatosis (median OS, 28 months) had significantly poorer survival compared to the control group (median OS, 95 months) [20]. Kobayashi et al. reported, in their 465 HCC patients undergoing hepatic resection, that Sa-O was an independent risk factor for OS (HR = 2.504, *p* = 0.005) and HCC recurrence (HR = 2.031, *p* = 0.006) after surgery [22]. Carias et al. reported that nonalcoholic steatohepatitis increases the risk of Sa-O six-fold in LC patients [23]. Hara et al. reported that the adverse prognostic impact of Sa-O was pronounced in LC patients with Child-Pugh A [24]. The diagnostic value of Sa-O is thus expected to increase in the future, and appropriate interventions in LC patients with Sa-O seem to be essential. However, to date, there are no interventional trials specific to cirrhotic patients with Sa-O.

### 7.1. Nutritional Intervention

The etiology of malnutrition in cirrhosis is diverse, including reduced food intake, malabsorption, and alteration in energy metabolism. In particular, as a result of impaired liver function for glucose metabolism, even after just one night of fasting, cirrhotic patients shift quickly to fat and protein catabolism, resulting in rapid muscle breakdown [83]. As described earlier, fasting for only 6–10 h in an LC patient is equivalent to two to three days of starvation in a healthy person [83]. In our 432 CLD patients undergoing indirect calorimetry, the prevalence of PEM significantly increased according to the LC status (2.8% in non-LC patients, 13.8% in Child-Pugh A, 52.5% in Child-Pugh B, and 76.9% in Child-Pugh C) [84]. To combat “accelerated starvation”, cirrhotic patients should avoid fasting for more than 6 h. Small frequent meals and late evening snacks that contain at least 50 g of complex carbohydrates, as well as protein sources, should be recommended in cirrhotic patients with PEM [85]. A recent meta-analysis reported that late evening snacks, such as enriched BCAAs, in cirrhotic patients can be a useful intervention for reversing anabolic resistance and sarcopenia [85]. However, there are few recommendations for nutritional management specific to cirrhotic patients with Sa-O. Nutritional therapy for cirrhotic patients with Sa-O should follow the general nutritional management of cirrhotic patients [86,87]. Nutritional care should be directed by a dietitian who is familiar with nutritional support for cirrhosis. The goal of nutritional management in these patients is to increase SMM and muscle strength. In LC patients with Sa-O, caution should be exercised for the risk of progression of sarcopenia with weight loss, and patients who are advised to lose weight should be monitored for changes in body composition and muscle strength or physical activities, using appropriate assessment tools [88]. Weight reduction by dietary restriction in obese patients is mainly the result of fat mass loss (about three quarters of the weight loss), while SMM loss accounts for about one quarter of the weight loss [89]. During dietary restriction, intramuscular fat loss can also occur. To decrease the adverse effects of SMM loss, a combination of energy restriction and exercise has been used as a treatment. This combination therapy results in a benefit to physical function, even in patients without an increase in muscle mass [90].

### 7.2. Exercise

Exercise as a strategy to prevent or treat obesity is often reported in the literature and has been studied extensively in interventional studies. Additionally, exercise has been widely recommended to improve SMM, muscle strength, and physical activities in older adults [91]. Many LC patients lead a sedentary lifestyle, which can be linked to Sa-O. Although reported in a small number of cases, the literature on physical activity in cirrhosis shows consistent benefits of exercise, including improved cardiorespiratory endurance, increased SMM, improved muscle strength and health-related quality of life, and improved portal hypertension [92,93,94,95,96]. However, there are no current recommendations of exercise specific to cirrhotic patients with Sa-O. Many reports regarding exercise for cirrhotic patients present the effects of exercise therapy for two to three months, but the long-term effects of exercise therapy in cirrhotic patients are also unknown [92,93,94,95,96]. In addition, data for the impact of exercise in patients with advanced LC status (i.e., presence of ascites) are currently lacking. Advanced LC patients, such as Child-Pugh B or C, have, in general, decreased exercise capacity [97]. LC patients with poor general condition should be given exercise intervention, with careful consideration for the risk of falling.

### 7.3. Pharmacological Therapies

Data-evaluating pharmacologic therapies specific to cirrhotic patients with Sa-O are also currently lacking. There are several promising agents, such as BCAA enriched snacks, l-carnitine, or rifaximin, that have serum ammonia lowering effects. BCAA also can improve insulin resistance in cirrhotic patients [98]. It has been suggested that lowering serum ammonia level may improve sarcopenia by inhibition of muscle protein breakdown in cirrhotic patients, but the effect of pharmacological interventions on Sa-O is unknown [99]. Further investigations will be needed to examine the impact of pharmacological interventions on Sa-O in cirrhotic patients.

## 8. Summary and Closing Remarks

In this review, we overviewed the relationship between cirrhosis and Sa-O pathologically and clinically. About three decades have passed since the proposal of sarcopenia, and about two decades have passed since the proposal of sarcopenic obesity. In 2016, sarcopenia assessment criteria specific to liver disease were proposed in Japan [100]. Sarcopenia and obesity are closely related. Liver cirrhosis can cause secondary sarcopenia, and obese CLD patients have been increasing in recent years. The pathogenesis of Sa-O in liver cirrhosis is multifactorial. There is a potential interplay among sarcopenia, obesity, myosteatosis and dysbiosis in cirrhotic patients. Insulin resistance can be one of key players in cirrhotic patients with Sa-O. Sa-O can be a poor prognostic factor in cirrhosis, as well as sarcopenia, but there are some problems, such as a lack of unified definition and established interventions for cirrhotic patients with Sa-O. Nutritional care should be directed by an expert dietitian in nutritional support for cirrhosis. In LC patients with Sa-O, caution should be paid for the risk of progression of sarcopenia with weight loss. LC patients with poor general condition should be given exercise intervention, with careful consideration for the risk of adverse events such as falling. Further novel evidence for Sa-O in LC is eagerly awaited.

## Figures and Tables

**Figure 1 ijms-22-01917-f001:**
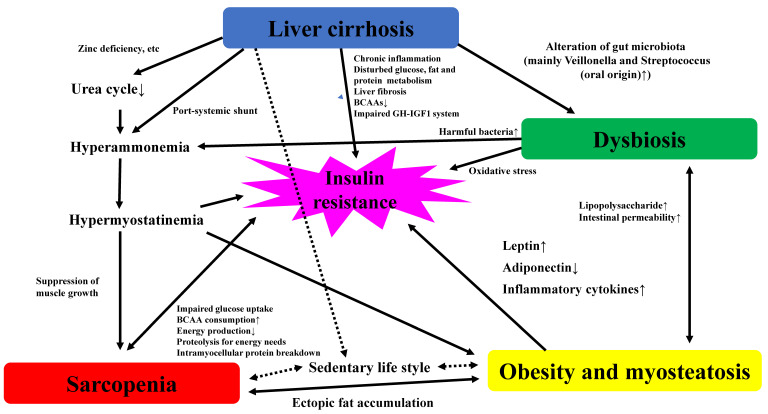
The potential interplay among patients with sarcopenia, liver cirrhosis, obesity, myosteatosis, and dysbiosis. BCAA, branched-chain amino acid; GH, growth hormone; IGF-1, insulin-like growth factor-1.

**Table 1 ijms-22-01917-t001:** Definition and prevalence of sarcopenic obesity (Sa-O) among studies.

Author (Country and Year)	Diagnostic Method (Skeletal Muscle)	Diagnostic Method (Fat Mass)	Definition (Sarcopenia)	Definition (Obesity)	Cutoff Point for Sarcopenia	Cutoff Point for Obesity	Prevalence of Sa-O (%)
Vidot H., et al. (Australia, 2019) [29]	CT (PMI, L3)‘	BMI	PMM/height^2^ (mm^2^/m^2^)	kg/m^2^	M: <545 F: <385	M: ≥30 F: ≥30	28%
Kroh, et al. (Germany, 2019) [27]	CT (L3 level)	Body fat percentage	SMM/height^2^ (cm^2^/m^2^)	%	M: <43 (BMI < 25) or <53 (BMI >25), F: <41	Top two quintiles	23%
Kobayashi, et al. (Japan, 2019) [22]	CT (L3 level)	CT (Umbilicus level)	SMM/height^2^ (cm^2^/m^2^)	cm^2^	M: <40.31 F: <30.88	M: ≥100 F: ≥100	7%
Kamo, et al. (Japan, 2019) [26]	CT (L3 level)	VFA or BMI	SMM/height^2^ (cm^2^/m^2^)	cm^2^ or kg/m^2^	M: <40.31 F: <30.88	VFA ≥ 100 or BMI ≥ 25	VFA: 3% BMI: 2%
Bering, et al. (Brazil, 2018) [28]	DXA and GS	Body fat percentage	SMM/height^2^ (cm^2^/m^2^) and kg	%	M: <7.46 and 30 (GS) F: <5.45 and 20 (GS)	M: ≥27 F: ≥37	14%
Hammad, et al. (Japan, 2017) [25]	CT (PMI, L3)	BMI	PMM/height^2^ (cm^2^/m^2^)	kg/m^2^	M: <6.36 F: <3.92	M: ≥25 F: ≥25	5%
Carias, et al. (USA, 2016) [23]	CT (L3-L4 level)	BMI	SMM/height^2^ (cm^2^/m^2^)	kg/m^2^	M: <52.4 F: <38.5	M: ≥30 F: ≥30	42%
Hara, et al. (Japan, 2016) [24]	BIA (ULM)	CT (Umbilicus level)	ULM/height^2^ (cm^2^/m^2^)	cm^2^	M: <1.7 F: <1.2	M: ≥100 F: ≥100	9%
Montano-Loza, et al. (Canada, 2016) [20]	CT (L3 level)	BMI MA at the L3 level	SMM/height^2^ (cm^2^/m^2^)	kg/m^2^ MA	M: <43 (BMI < 25) or <53 (BMI > 25), F: <41	M and F: BMI ≥ 25 or MA < 33 HU	20%
Kaibori, et al. (Japan, 2015) [21]	CT (L3 level)	CT (multifidus muscle at the umbilicus level)	SMM/height^2^ (cm^2^/m^2^)	IMAC	M: <44 F: <38	M: −0.44 F: −0.31	NA

CT; computed tomography, BMI; body mass index, PMI; psoas muscle index, PMM; psoas muscle mass, M; male, F; female, SMM; skeletal muscle mass, VFA; visceral fat area, DXA; dual-energy X-ray absorptiometry, GS; grip strength, BIA; bioelectrical impedance analysis, ULM; upper limb mass, MA; muscle attenuation. IMAC; intramuscular adipose content, HU; Hounsfield Unit, NA; not assessed.

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
