# Peer review of "Sarcopenic Obesity in Liver Cirrhosis: Possible Mechanism and Clinical Impact"

_ijms, 2021, doi:10.3390/ijms22041917_

Round 1
Reviewer 1 Report
What about …IFN-s-Dependent Muscle Decay as per…Mediators Inflamm. 2013; 2013: 171437, mainly in viral cirrhosis without or with antiviral therapy based on IFNs.
Authors should expand this topic to give readers a deeper knowledge of the issue.
Authors should further comment on the main role of adiponectin in liver cirrhosis as per …Ann Hepatol. 2018 Mar 1; 17(2):286-299.
Authors should expand this topic to give readers a wider view.
About the main mechanisms of adiponectin in liver cirrhosis following NASH authors could refer to …Diabetes Metab. 2014;40:95–107 and World J Gastroenterol. 2013 Feb 14;19(6):802-12.
Author Response
Thank you very much for the opportunity to submit our revised manuscript to your highly respected journal. We would like to thank the editors and the reviewers for their thoughtful comments and are convinced that we have substantially improved our manuscript by carefully revising it according to their suggestions. Please find a detailed point-by-point response to the reviewers’ comments attached to this letter.
Response to reviewer 1
# What about IFN-s-Dependent Muscle Decay as per…Mediators Inflamm. 2013; 2013: 171437, mainly in viral cirrhosis without or with antiviral therapy based on IFNs.
Our response:
In the revised section 4, we added the following sentence:
TNF-α and IFN-s-dependent muscle decline is linked to NF-κB pathway.
In addition, we quoted the article you recommended.
# Authors should expand this topic to give readers a deeper knowledge of the issue. Authors should expand this topic to give readers a wider view.
Our response:
In this article, we aimed to review the relationship between cirrhosis and sarcopenic obesity pathologically and clinically, and summarized the current knowledge and problems in the final section. We appreciate your warm understanding.
# Authors should further comment on the main role of adiponectin in liver cirrhosis as per …Ann Hepatol. 2018 Mar 1; 17(2):286-299.
Our response:
In the revised section 4, we added the following sentence:
Adiponectin exerts anti-fibrotic and anti-inflammatory effect in LC.
In addition, we quoted the article you recommended.
# About the main mechanisms of adiponectin in liver cirrhosis following NASH authors could refer to …Diabetes Metab. 2014;40:95–107 and World J Gastroenterol. 2013 Feb 14;19(6):802-12.
Our response:
In the revised section 4, we added the following sentence:
Especially in patients with NAFLD, the protective effects of adiponectin for hepatocytes have been widely examined.
In addition, we quoted two articles you recommended.
Response to reviewer 2
# This manuscript provides a well written and detailed review of the possible mechanisms and the clinical impact of sarcopenic obesity in cirrhosis. The most pertinent literature has been properly reviewed and commented, and the topic is of high relevance for people working on the field. I recommend acceptance of the manuscript as it is. Some minor editing and typos need to be corrected (very few).
Our response:
Thank you for your comments. We read the whole paper carefully, and corrected typo errors.
We hope that this revised manuscript is now suitable for publication in IJMS.
Thank you for time and attention
Yours sincerely,
Hiroki Nishikawa
Hyogo College of Medicine, Japan

Reviewer 2 Report
This manuscript provides a well written and detailed review of the possible mechanisms and the clinical impact of sarcopenic obesity in cirrhosis. The most pertinent literature has been properly reviewed and commented, and the topic is of high relevance for people working on the field.
I recommend acceptance of the manuscript as it is.
Some minor editing and typos need to be corrected (very few).
Author Response
Thank you very much for the opportunity to submit our revised manuscript to your highly respected journal. We would like to thank the editors and the reviewers for their thoughtful comments and are convinced that we have substantially improved our manuscript by carefully revising it according to their suggestions. Please find a detailed point-by-point response to the reviewers’ comments attached to this letter.
Response to reviewer 1
# What about IFN-s-Dependent Muscle Decay as per…Mediators Inflamm. 2013; 2013: 171437, mainly in viral cirrhosis without or with antiviral therapy based on IFNs.
Our response:
In the revised section 4, we added the following sentence:
TNF-α and IFN-s-dependent muscle decline is linked to NF-κB pathway.
In addition, we quoted the article you recommended.
# Authors should expand this topic to give readers a deeper knowledge of the issue. Authors should expand this topic to give readers a wider view.
Our response:
In this article, we aimed to review the relationship between cirrhosis and sarcopenic obesity pathologically and clinically, and summarized the current knowledge and problems in the final section. We appreciate your warm understanding.
# Authors should further comment on the main role of adiponectin in liver cirrhosis as per …Ann Hepatol. 2018 Mar 1; 17(2):286-299.
Our response:
In the revised section 4, we added the following sentence:
Adiponectin exerts anti-fibrotic and anti-inflammatory effect in LC.
In addition, we quoted the article you recommended.
# About the main mechanisms of adiponectin in liver cirrhosis following NASH authors could refer to …Diabetes Metab. 2014;40:95–107 and World J Gastroenterol. 2013 Feb 14;19(6):802-12.
Our response:
In the revised section 4, we added the following sentence:
Especially in patients with NAFLD, the protective effects of adiponectin for hepatocytes have been widely examined.
In addition, we quoted two articles you recommended.
Response to reviewer 2
# This manuscript provides a well written and detailed review of the possible mechanisms and the clinical impact of sarcopenic obesity in cirrhosis. The most pertinent literature has been properly reviewed and commented, and the topic is of high relevance for people working on the field. I recommend acceptance of the manuscript as it is. Some minor editing and typos need to be corrected (very few).
Our response:
Thank you for your comments. We read the whole paper carefully, and corrected typo errors.
Round 2
Reviewer 1 Report
Authors correctly answered comments